# Engineering and Preclinical Evaluation of Western Reserve Oncolytic Vaccinia Virus Expressing A167Y Mutant Herpes Simplex Virus Thymidine Kinase

**DOI:** 10.3390/biomedicines8100426

**Published:** 2020-10-16

**Authors:** S. M. Bakhtiar UL Islam, Young Mi Hong, Mefotse saha Cyrelle Ornella, Daniel Ngabire, Hyunjung Jang, Euna Cho, Eung-Kyun Kim, Jessye Jin Joo Hale, Cy Hyun Kim, Soon Cheol Ahn, Mong Cho, Tae-Ho Hwang

**Affiliations:** 1Medical Research Center, School of Medicine, Pusan National University, Yangsan 50612, Korea; smbakhtiar@pusan.ac.kr (S.M.B.U.I.); krxsorne@gmail.com (M.s.C.O.); danyngabire@gmail.com (D.N.); biochiphyun@gmail.com (H.J.); 2Department of Microbiology and Immunology, School of Medicine, Pusan National University, Yangsan 50612, Korea; ahnsc@pusan.ac.kr; 3Liver Center, Pusan National University Yangsan Hospital, Department of Internal Medicine, School of Medicine, Pusan National University, Yangsan 50612, Korea; 00gurum@hanmail.net (Y.M.H.); mcho@pusan.ac.kr (M.C.); 4Bionoxx Inc., Parkview Tower #1905, 248 Jeongjail-ro, Bundang-gu, Seongnam-si, Gyeonggi-do 13554, Korea; euna.cho@bionoxx.com (E.C.); ekkim@bionoxx.com (E.-K.K.); jinjoo.hale@bionoxx.com (J.J.J.H.); chkim@bionoxx.com (C.H.K.); 5Department of Pharmacology, School of Medicine, Pusan National University, Yangsan 50612, Korea

**Keywords:** oncolytic viruses, western reserve vaccinia virus, herpes simplex virus, *thymidine kinase*, A167Y, ganciclovir

## Abstract

Viral replication of *thymidine kinase* deleted (*tk*^−^) vaccinia virus (VV) is attenuated in resting normal cells, enabling cancer selectivity, however, replication potency of VV*-tk^−^* appears to be diminished in cancer cells. Previously, we found that wild-type herpes simplex virus (HSV*)-tk* (*HSV-tk*) disappeared in most of the recombinant VV after multiple screenings, and only a few recombinant VV containing naturally mutated *HSV-tk* remained stable. In this study, VV*-tk* of western reserve (WR) VV was replaced by A167Y mutated *HSV-tk* (*HSV-tk_418m_*), to alter nucleoside selectivity from broad spectrum to purine exclusive selectivity. WOTS-418 remained stable after numerous passages. WOTS-418 replication was significantly attenuated in normal cells, but cytotoxicity was almost similar to that of wild type WR VV in cancer cells. WOTS-418 showed no lethality following a 5 × 10^8^ PFU intranasal injection, contrasting WR VV, which showed 100% lethality at 1 × 10^5^ PFU. Additionally, ganciclovir (GCV) but not BvdU inhibited WOTS-418 replication, confirming specificity to purine nucleoside analogs. The potency of WOTS-418 replication inhibition by GCV was > 10-fold higher than that of our previous truncated *HSV-tk* recombinant OTS-412. Overall, WOTS-418 demonstrated robust oncolytic efficacy and pharmacological safety which may delegate it as a candidate for future clinical use in OV therapy.

## 1. Introduction

Vaccinia virus (VV) is one of the more preferred backbones for oncolytic virus (OV) engineering based on its long history of use as a vaccine for smallpox. In clinical trials, after systemic administration of oncolytic vaccinia virus (OVV), delayed re-emergence of circulating OVV appeared in the blood on day 3–7 (as a peak), consistent with viral replication [1]. Persistent presence of OVV can provide beneficial antitumor effects [2,3], but it can lead to safety concerns, which were also raised by the US FDA in the guidance of “Preclinical Assessment of Investigational Cellular and Gene Therapy Products” when dealing with replication-competent oncolytic viruses (November 2013). 

*Thymidine kinase* (*tk^−^*) deleted VV have been developed to increase cancer selectivity using methods to substantially attenuate viral replication in normal cells, but these methods can also significantly decrease cytotoxic potency in cancer cells, which may lead to relatively poor clinical outcomes [4,5]. Generally, *tk* (human cytosolic *tk* (TK1) or viral *tk*) uses ATP as a phosphate donor to deoxyribonucleosides during DNA synthesis [6]. Nucleoside selectivity of these different TKs is determined by the nucleoside binding site (NBS) encoded in the *tk* gene [7]. The NBS coding regions of *VV-tk* and human TK1 (*H-tk*) share significant homology, which may provide a basis for the cancer selectivity of *tk* deleted VV (*VV-tk^−^*), but there is no significant homology shared with the NBS of herpes simplex virus *tk* (*HSV-tk*) [8]. Efficient DNA synthesis is determined by nucleoside selectivity as well as the coordinated increase (10–20 fold) of other kinases—such as thymidylate synthase [9], ribonucleotide diphosphate reductase [10], and dihydrofolate reductase [11]—in the G1/S phase of cell cycle. This orchestrated kinase activity plays slightly different roles in each of the following: viral DNA replication, cancer cell proliferation, and proliferation in normal cells, which may be further subdivided by cell type, where activity may vary between cell types such as bone marrow cells and gastrointestinal epithelial cells. 

Over the past few decades, nucleoside analogs have been developed as antiviral agents, and purine and pyrimidine analogs have shown differential anticancer effects, as well as other side effects, suggesting that either purine or pyrimidine nucleoside phosphorylation may be a limiting factor for DNA synthesis depending on tumor type [7,12,13,14,15,16]. While the NBS of *HSV-tk* recognizes both purine and pyrimidine nucleoside analogs, those of *H-tk* and *VV-tk* recognize pyrimidine, but not purine nucleosides. This nucleoside metabolism of *HSV-tk* can be altered in order to limit binding to specific nucleotide analogs via a single amino acid mutation in or around NBS of *HSV-tk* [17,18,19]. Among reported mutations, we noted the A167Y(pAla167Tyr) of HSV-*tk* mutation (designated as a *HSVtk_418m_* in this study) completely abolished pyrimidine nucleoside kinase activity; however, purine nucleoside kinase activity was maintained [18,19]. After A167Y substitution, GCV phosphorylation by the mutant *HSV-tk* was not changed, even in the presence of high level deoxythymidine (dTh) [19]. This outcome differs from the substantial and preferential phosphorylation of dTh by wild type *HSV-tk,* meaning that the mutant had significantly increased GCV potency by blocking TK activity. In normal cells, there is low level of resting deoxynucleotide (dNTP); however, in cancer cells high TK activity results in high levels of dNTPs, especially pyrimidines. 

Taken together, we hypothesized that *VV-tk^−^*/*HSV-tk_418m_* armed recombinant OVV, WOTS-418, which selectively phosphorylates purine nucleosides and its analogs, would be efficiently controllable by GCV treatment. As opposed to WR VV, the replication potency of *VV-tk^−^*/*HSVtk_418m_* expressing recombinant OVV would be attenuated in normal cells but maintained in some cancer cells. Furthermore, considering that GCV phosphorylation by *HSV-tk_418m_* is higher than by wild type *HSV-tk*, aberrant viral replication would be controllable by pharmacological intervention, allowing for the use of the more aggressive WR VV back bone in OV therapy. This pharmacological activity substantially contrasts from our previous OTS-412 studies where *VV-tk* gene was replaced with 3′ end truncated *HSV-tk* [20]. Homologous recombination (HR) lysates from this previous study were cultured in 143B cells under *tk* negative pressure in the presence of bromodeoxyuridine (BrdU) and resulted in the generation of multiple mutants of *HSV-tk* expressing OVV. Following this, OTS-412 (armed with a truncated HSV-*tk*) was selected out of 120 different single clones after pharmacological characterization and showed over 50% virus replication control in vivo [20]. 

Here, we showed that WOTS-418, a new *HSV-tk_418m_* armed *VV-tk* deleted OVV, has improved tumor selectivity, robust oncolytic capacity in multiple human solid tumors, and that GCV treatment is substantially potent in inhibition of undesirable viral replication (>10 fold) both in vitro and in vivo.

## 2. Materials and Methods

### 2.1. Cell Lines and Virus

Human osteosarcoma (143B and U-2 OS), human cervical adenocarcinoma (HeLa and HeLa S3), human epithelial lung carcinoma (A549), murine colorectal cancer cell (CT-26.WT), and murine breast carcinoma (4T1) cell lines were purchased from the American Type Culture Collection (ATCC; Manassas, VA, USA). Human lung carcinoma (NCI-H460), human normal lung fibroblast (MRC-5), human kidney carcinoma (A-498 and Caki-1), human colorectal adenocarcinoma (HCT 116), human metastatic breast carcinoma (MDA-MB-231 and MCF7), and murine renal adenocarcinoma (Renca) were obtained from the Korean Cell Line Bank (KCLB, Chongno-gu, Seoul, Korea). All cell lines were maintained with ATCC and KCLB recommended media, respectively, and supplemented with 10% fetal bovine serum (FBS). The NIH TC-adapted WR strain of VV (VR-1354, ATCC; Manassas, VA, USA) was purchased from the ATCC, amplified in HeLa or HeLa S3 cells, and quantified using a vaccinia virus titration protocol. WR-GFP is a *tk* deleted and green fluorescence protein armed *tk-*deleted WR VV obtained from Professor Kim Jae-Ho, Pusan National University, South Korea. All cell lines and viruses were cultured under appropriate conditions and were routinely tested for sterility and mycoplasma contamination using MycoAlert assay kit (Lonza, Rockland, ME, USA).

### 2.2. WOTS-418 Engineering and Characterization

To engineer WOTS-418, *HSV-tk* gene (NCBI GenBank: J02224.1) coding sequence was redesigned by a single amino acid A167Y (p.Ala167Tyr) modification, as reported earlier, by altering the nucleotides c.499G > T and c.500C > A, respectively. However, additional alterations were performed via substitution of nucleotides c.432G > T, c.435G > T, and c.552C > T, respectively, without altering the native amino acid sequences. This new version of the inserted *HSV-tk* was referred to as *HSV-tk_418m_* and contains the early transcription termination sequence (5′-TTTTTAT-3′) at the downstream of the modified the *HSV-tk* coding sequence. Two restriction sites, KpnI (5′-GGTACC-3′) and NheI (5′-GCTAGC-3′), were added at the 5′ and 3′-end, respectively, for the purpose of future cloning inside the pOTS shuttle plasmid, as described earlier in OTS-412 engineering. After synthesis of *HSV-tk_418m_* gene cassette, the inserted fragment was cloned into the pUC57 cloning plasmid and the DNA sequence was then confirmed by Cosmogenetech Inc, Seoul, South Korea. Next, the newly synthesized plasmid (pINSER1) and pOTS were digested with the restriction enzymes NheI (NEB, Ipswich, MA, USA) and KpnI (NEB, Ipswich, MA, USA) to remove *HSV-tk* from pOTS shuttle vector. The final shuttle plasmid, pOTS-418 was constructed following standard cloning procedure (gel extraction, ligation, and transformation) and was further characterized by restriction enzymes (NheI and KpnI) digest analysis as well as DNA sequencing (Cosmogenetech Inc, Seoul, South Korea). Thus, the shuttle plasmid pOTS-418 retained almost all features of the pOTS shuttle except the *HSV-tk*, which was replaced by the *HSV-tk_418m_*. After this shuttle plasmid construction, HR experiment was performed in HeLa cells for 48 h, as described previously [20]. HR was confirmed in both supernatant and cell pellet by firefly luciferase assay (E1500, Promega, Madison, WI, USA) according to the instruction manual. Next, the *firefly luciferase* positive recombinant virus mixture was cultured in 143B cell line (maintained with 15 µg/mL BrdU) to create a TK negative selection pressure, which was performed three times, as described earlier. During the first round of screening, approximately sixty-five (65) distant single plaques were isolated and screened for *firefly luciferase* gene expression. Among those, five luciferase expressing single plaques (based on expression level) were evaluated for luciferase gene expression (supernatant), HSV-tk DNA insertion confirmation by polymerase chain reaction (PCR), and *HSV-tk* gene expression, by western blotting. One of these five plaques was selected for a second round TK negative selection pressure, and finally a distant single and pure single plaque was isolated (from the third round *tk* negative selection pressure) which was ultimately referred to as WOTS-418. This final plaque underwent single plaque isolation procedure one more time to confirm the 100% purity of WOTS-418. 

To characterize the WOTS-418, *luciferase* expression, *HSV-tk* expression, gene sequencing, HindIII digestion pattern, and purine/pyrimidine analog antiviral potency was evaluated. A549 cells (5 × 10^5^ cells/well) were cultured in 6-well plates overnight and were infected the next day with WR VV and WOTS-418 at 0.1 PFU/cell for 24 h. 20 µL supernatant was used to analyze *firefly luciferase* gene expression, and whole cell pellets were used to confirm the *HSV-tk* gene expression by western blotting. To sequence the transgene of WOTS-418, total DNA was extracted using a DNeasy blood and tissue DNA extraction kit (69504, Qiagen, CA, USA) and DNA concentration was determined using TECAN NanoQuant plate and nucleic acid quantification system (TECAN 2000, Männedorf, Switzerland). HSV1-TK SF primer and HSV1-TK SR were used to amplify the target *HSV-tk* region under the following PCR conditions: initial denaturation at 95 °C for 2 min; followed by 30 cycles of denaturation at 95 °C for 45 s, annealing at 58 °C for 45 s, elongation at 72 °C for 90 s with the final step at 72 °C for 5 min using HelixAmp™ Ready-2x-Go PCR kit (PMD008L, Nanohelix, Daejeon, Korea). The PCR product was sent to Macrogen Inc, Seoul, South Korea for sequencing. The sequencing primers used in these experiments were SEQ 1, SEQ 2, SEQ 3, and SEQ 4, respectively. All primer sequences are listed in Table 1. We also performed HindIII restriction enzyme digestion analysis for WR VV and WOTS-418 as described later in the restriction enzyme digest analysis methods. Finally, to evaluate the purine/pyrimidine antiviral effect, 143B cells (TK-) were seeded in six well plates (5 × 10^5^ cells/well) and incubated overnight, and then infected with 0.1 PFU/cell WOTS-418, with or without purine analogs (ACV and GCV) and pyrimidine analog (BvdU) at 100 µM concentration. Wild type HSV-1 virus was included as a positive control, following the same experimental set-up. After 72 h, cells and supernatant were harvested and processed by three cycles of freezing–thawing and then sonication (100% power, 45 s, 3 times with 60 s interval). Each sample (HSV-1 and WOTS-418) was serially diluted 10-fold and used to reinfect HeLa cells, which had been pre-seeded in 96-well plates (1 × 10^4^ cells/well). After 72 h, cell viability was measured using a cell counting kit (CCK-8, Dojindo, Kumamoto, Japan), and quantified using the microplate spectrophotometer (Spark, Tecan 2000, Männedorf, Switzerland) at 450 nm. Next, WOTS-418 samples were further evaluated by vaccinia virus titration or plaque assay. 

To evaluate tumor selectivity, WOTS-418 was used to infect normal human cell line and the results were compared to those of WR VV. MRC-5 cells were seeded in 96 well plates at 3 × 10^4^ cells/well and infected with WR VV or WOTS-418 (0.1 PFU/cell and 1 PFU/cell). After 48 h incubation, phase-contrast photomicrographs (Magnification: 100×) were taken using Incucyte S3 live-cell imaging system (Essen BioScience, Ann Arbor, MI, USA) to evaluate the cytopathic effect; then, cell viability was measured by CCK-8 kit (Dojindo, Kumamoto, Japan). Furthermore, to compare the viral replication of WOTS-418 in normal human cells and cancer cells, MRC-5 and HeLa cells were seeded in 96-well plates at 3 × 10^4^ cells/well and infected with WOTS-418 (1 PFU/cell). At 48 h after infection, 20 μL supernatant was aspirated to measure the *firefly luciferase* expression (Spark, Tecan 2000, Männedorf, Switzerland) using a luciferase assay kit (E1500, Promega, WI, USA).

For preclinical in vitro and in vivo testing, all recombinant viruses were amplified, aliquoted, quantified by titration, and stored at −80 °C until use.

### 2.3. Virus Titration by Plaque Assay

For vaccinia virus titration, plaque assay was performed. In brief, 5 × 10^5^ U-2 OS cells/well were seeded in 6-well plates, and the next day, serially diluted virus in 2% DMEM (infection media) was used to infect the cells for a duration of 2 h. Next, infection media was aspirated and replaced with 1.5% carboxymethylcellulose and 2% DMEM. Plates were incubated at 37 °C with 5% CO_2_ for next 72 h. Viral plaques that were visible after staining with 0.5% crystal violet dissolved in 20% ethanol, were manually counted (counting range 20–200), and virus titration was calculated according to the formula: Virus titration (PFU/mL) = Average virus plaque numbers / Dilution factor × Infection volume (mL). 

### 2.4. Virus Purification 

HeLa or HeLa S3 cells were infected with recombinant viruses (0.01 pfu/cell) for 72 h. Cell pellets were collected and suspended in 10mM Tris-HCl buffer (pH 9.0). After homogenization, samples were put on top of the 36% sucrose (Samchun, Busan, Korea) solution prepared in 10 mM Tris-HCl buffer (pH 9.0) and high speed centrifuged at 16,490× *g* rpm for 80 min. The resulting pellet was re-suspended in 1mM Tris-HCl buffer (pH 9.0), sonicated (100% power for 45 s with 3 times interval for 1 min), aliquoted, and stored at −80.0 °C. 

### 2.5. Firefly Luciferase Assay

A549 cells were seeded (4 × 10^5^ cells/well) in 6-well plate, and after 24 h, infected with 0.1 pfu/cell of each of the viruses. 24 h post infection (*p.i.*), supernatant and cell pellet were harvested and used to detect *firefly luciferase* expression following the manufacturer’s protocol (E1500, Promega, Madison, WI, USA). 

### 2.6. PCR

A549 cells were infected with 0.1 pfu/cell of the viruses for 24 h. Total DNA was extracted by DNeasy Blood & Tissue kit according to the manufacturer’s instructions (Qiagen 69504, Valencia, CA, USA). DNA concentration was measured with a Nano instrument (Nanodrop2000, Thermo, Waltham, MA, USA). HSV-TK DNA region was amplified with HelixAmp™ Ready-2x-Go (Taq-Plus) PCR kit (Nano Helix PDM008L, Daejeon, Korea) using the following PCR condition: 95 °C for 5 min; followed by 27 cycles consisting of: 95 °C for 45 s, 55 °C for 45 s, and 72 °C for 90 s; followed by a 5 min incubation at 72 °C. The primer used to amplify the *HSV-tk* is listed in Table 1. The PCR product image was captured after running in 1% agarose gel (Promega V3125, Madison, WI, USA). 

### 2.7. Western Blotting

For western blotting, whole cell lysates were prepared using RIPA lysis buffer (GenDEPOT, Katy, TX, USA). Lysates were then centrifuged at 14,000× *g* for 10 min at 4 °C. Supernatants were loaded to SDS-PAGE (4–15%) gradient gels and transferred to PVDF membranes (Millipore, USA cat. no. IPVH00010). Membranes were incubated overnight at 4 °C with primary antibody, herpes simplex virus type 1 thymidine kinase (SC-28037, Santa Cruz, Dallas, TX, USA), and washed three times in PBS with 0.1% Tween 20 prior to 1 h incubation with secondary antibody, anti-Goat IgG (no. A50-101P, Bethyl, Montgomery, TX, USA), at room temperature. Proteins were then detected using chemi-doc system (no. CAS400SM, Davinch-K, Gangnam-gu, Seoul, Korea).

### 2.8. Restriction Enzyme Digest Analysis

U2-OS cells were infected with viruses at 0.1 PFU/cell for 48 h. To prepare virus stock, cellular DNA was removed by treatment with cytoplasmic lysis buffer (10 mM Tris-HCl, pH8.0, 10 mM KCl, 5 mM Na_2_EDTA). Next, the stock virus was homogenized using 1 mL syringe containing lysis buffer (54% sucrose, 2-mercaptoethanol, proteinase-K, 10% SDS, and 5M NaCl). Each virus’ DNA was extracted using QIAGEN Gel Extraction Kit (QIAquick Gel Extraction Kit, Qiagen, Hilden, Germany) and DNA concentration was quantified the using Nano Quant module of TECAN instrument (NanoQuant Plate, TECAN, Männedorf, Switzerland). Next, for the restriction digest experiment, 2.5 μg DNA from each virus was digested with HindIII-HF restriction enzyme (R3104S, NEB, Ipswich, MA, USA) in a 37 °C water bath for 12 h. The digested DNA was run on a 0.6% agarose gel at 50 V for 220 min. Images were captured while using a chemiluminescent imaging system using Gel-doc option (CAS-400SM, Davinch-K, Seoul, Korea).

### 2.9. Gene Sequencing

To sequence the HSV-*tk* transgene in WOTS-418, total DNA was extracted using a DNeasy blood and tissue DNA extraction kit (69504, Qiagen) and DNA concentration was determined using TECAN NanoQuant plate and nucleic acid quantification system (TECAN 2000, Männedorf, Switzerland). HSV1-TK SF forward primer and HSV1-TK SR reverse primer were used to amplify the target *HSV-tk* region by using the following polymerase chain reaction (PCR) conditions: initial denaturation at 95 °C for 2 min; followed by 30 cycles of denaturation at 95 °C for 45 s, annealing at 58 °C for 45 s, elongation at 72 °C for 90 s, and the final step at 72 °C for 5 min using HelixAmp™ Ready-2x-Go PCR kit (PMD008L, Nanohelix, Daejeon, Korea). The PCR product was sent for sequencing to either Cosmogenetech Inc, Seoul, South Korea or Macrogen Inc, Seoul, South Korea. The sequencing primers used in these experiments were SEQ 1, SEQ 2, SEQ 3, and SEQ 4, respectively, as listed in Table 1.

### 2.10. Cytotoxicity Assay (In Vitro)

All cytotoxicity assays were performed using cell counting kit (Dojindo, Kumamoto, Japan) according to the manufacturer’s protocol, and analyzed by TECAN 2000 spectrophotometer (Spark, Männedorf, Swiss) at 450 nm. The second method of cytotoxicity observation was using Incucyte S3 live-cell imaging (Essen BioScience, Ann Arbor, MI, USA) analysis and looking at time dependent cell confluency via phase contrast image analysis. 

### 2.11. qPCR

Total DNA extraction was performed according to the QIAamp DNA Mini Blood Kit (51106, QIAgen, Valencia, CA, USA). Quantification of viral DNA was performed using 2× TaqMan Universal PCR Master Mix (4304437, Applied Biosystem, Foster City, USA) and QuantStudio 5 Real-Time PCR instrument (Thermo Fisher Scientific, Waltham, MA, USA). Each sample was analyzed in duplicate or triplicate for fast quantitative-PCR analysis and specific probe and primers (E9L Forward and E9L Reverse) were used. For thermal cycling, PCR instrument was set up to amplify the vaccinia virus specific E9L gene under the following conditions: pre-denaturation at 50 °C for 2 min, denaturation at 95 °C for 10 min, followed by 40 cycles of denaturation at 95 °C for 15 s for annealing, and extension at 60 °C for 90 s. For standard curve, E9L plasmid was diluted (1 × 10^7^ copies to 2.5 copies/5 μL). The number of vaccinia virus particles was calculated by the correlation between threshold level (Ct) level and E9L copies. Primers used in this experiment are listed in the Table 1.

### 2.12. Animal Models 

Athymic and syngeneic female or male mice were purchased from Orient Bio Inc., South Korea and KOATECH Inc., South Korea. After arrival, mice were acclimatized to the new environment for a week before starting any drug treatments or tumor formation. All animals were handled in compliance with Institutional Animal Care and Use Committee (IACUC) guidelines and each animal study was approved by the Pusan National University’s IACUC under the study numbers PNU-2019-2131 (16 January 2019) and PNU-2019-2339 (23 July 2019).

### 2.13. Statistical Analysis 

All of the statistical analyses were performed using Prism 8 (GraphPad software, La Jolla, CA, USA). Unpaired two-tailed *t*-tests were used to determine the experimental significance between the groups. Statistical significance was indicated as *p*-values < 0.05 unless otherwise stated in the figure legend.

## 3. Results

In this study, wild type *HSV-tk* was modified for purine selectivity and gene stability and incorporated in WR VV via deletion of native *tk* gene using homologous recombination. This new recombinant OVV, WOTS-418, was characterized in vitro and evaluated in preclinical animal models. The antitumor potency was investigated in two human solid tumor models and GCV mediated viral replication control was assessed both in vitro and in vivo. 

### 3.1. Engineering and Characterization of WOTS-418

The comparison of whole protein sequence and NBS region of H-*tk,* WR *VV-tk*, and *HSV-tk* is shown in the Appendix B in Figure A1. H-*tk* and WR *VV-tk* showed remarkable homology in the NBS region whereas H-*tk* and *HSV-tk* or WR *VV-tk* and *HSV-tk* did not showed significant homology indicating a basis for the cancer selectivity of *tk* deleted WR VV. 

To this end, we engineered the first replication-controlled OVV, OTS-412, armed with truncated *HSV-tk*. During, OTS-412 engineering, though our initial attempt was to incorporate the full *HSV-tk* in a VV backbone, instead we obtained a truncated version of *HSV-tk* armed VV (OTS-412) and few other mutants. Those mutants *HSV-tk* DNA sequences were further evaluated to figure out the frequency and location of genetic changes. In brief, after HR, more than 1500 *firefly luciferase* positive single plaques were isolated from HR lysate. Those plaques were used to re-infect 143B TK- cells (maintained with BrdU) and further screened for *firefly luciferase* gene expression. During this experiment, two distinct phenomena were observed. More than 99.8% of isolated single plaques showed unstable *luciferase* gene expression in presence or even in the absence of BrdU selection pressure. Those plaques were further screened by western blotting to check *HSV-tk* gene expression. None of those isolates showed *HSV-tk* gene expression when evaluated by western blotting (Figure 1A, top panel). Only a few plaques (less than 0.2%) showed variable yet stable *firefly luciferase* positive signal under BrdU selection pressure. *HSV-tk* gene expression analysis by western blotting showed that none of those plaques contained full length *HSV-tk,* but rather expressed one of the three distinct types of truncated *HSV-tk* bands which were 36.1 kDa (OTS-412), 24.5 kDa (OTS-C1), and 19.7 kDa (OTS-C2, respectively (Figure 1A, middle panel)) as reported earlier [20].

To avoid the genetic and functional conflicts associated with full length *HSV-tk* incorporation in VV backbone, we sought to design a modified *HSV-tk_418m_* gene cassette through substitution of nucleotide-G and nucleotide-C with the nucleotide-T at the selected ‘hot spot’ mutation regions, leaving the original amino acid sequences intact along with A167Y modification (Figure 1B). With this newly designed modified *HSV-tk_418m_,* we engineered the second version of replication-controllable OVV (WOTS-418 and OTS-418) using the WR VV strain and Wyeth VV strain, respectively (this paper will focus on WOTS-418). In this report, we will describe a new *tk* deleted recombinant WR VV armed with *HSV-tk_418m_* and *firefly luciferase*. In brief, after transfection of pOTS-418 shuttle plasmid in WR VV infected HeLa cells, the successful HR was initially confirmed by *luciferase* expression (Figure 1A, bottom panel). Next, during the first round of screening under BrdU selection pressure, 65 single plaques were isolated and screened for *luciferase* gene expression. In the first round of single plaque selection, all the isolated single plaques showed positive signal for *luciferase* gene expression (Figure 1A, bottom panel). Among those *luciferase* positive isolated single plaques, the five-*luciferase* gene expressing single plaques were further evaluated for the molecular detection of HSV-tk DNA by PCR and HSV-*tk* gene expression by western blotting. All five plaques showed an identical HSV-tk DNA band and HSV-*tk* gene expression when compared with the positive control pOTS-418 (PCR positive control, product size 1284 bp) and *HSV-tk* (approximately 41.0 kDa) of wild type HSV-1 virus. One of these single plaques was selected for two more rounds (second and third) of screening under the BrdU TK- selection pressure and finally a single distant plaque was isolated, which was named WOTS-418.

This new recombinant OVV, WOTS-418 was further characterized by investigating *luciferase* gene expression, HSV-*tk* DNA band size by PCR, *HSV-tk* gene expression by western blotting, purine/pyrimidine analog antiviral potency assay, HindIII restriction digestion assay and DNA sequencing. During this WOTS-418 characterization experiment, we found strong *luciferase* gene expression in the supernatant of WOTS-418 infected A549 cells. PCR experiments reconfirmed that WOTS-418 had an identical DNA band (1,284 bp) to the positive control shuttle plasmid pOTS-418 and the western blotting experiment showed the full-length *HSV-tk_418m_* gene band (approximately 41.0 kDa) was similar to the positive control *HSV-tk* from HSV-1 virus (Figure 1A, bottom panel). DNA sequence analysis of *HSV-tk_418m_* from WOTS-418 indicated the successful incorporation of the desired nucleotide modification, which was identical to the *HSV-tk_418m_* DNA sequence from the pOTS-418 shuttle plasmid. Location of the gene modifications and sequence confirmation after engineering is shown in the Figure 1B, middle panel. Primers used in this study are listed in Table 1.

Furthermore, *HSV-tk_418m_* gene expression stability was monitored up to the 10th amplification by western blotting and showed an identical band to the HSV-*tk* (positive control) and the *HSV-tk_418m_* band from the initial (first) amplification of WOTS-418 (Figure 1C). HindIII restriction digest experiment showed that the pattern of DNA band was altered after incorporation of transgenes (*HSV-tk_418m_* and *firefly luciferase*) when compared to the wild type WR VV DNA. This digestion pattern was also stable up to the 10th amplification. Mismatched DNA fragments are indicated by blue arrows (Figure 1D). 

After confirmation of successful WOTS-418 engineering and characterization, GCV mediated viral replication inhibition was measured in OTS-412 and WOTS-418. In this study, GCV treatment resulted in significant reduction of WOTS-418 virus particles (vp) copies (fold inhibition) in NCI-H460 (*p* = 0.016) and A549 (*p* < 0.001) cell lines compared to OTS-412, indicating that the *HSV-tk_418m_* from WOTS-418 seems to be stronger and more functional than the truncated *HSV-tk* from OTS-412 (Figure 1E). 

To understand the impact of the transgenes of WOTS-418 on cancer cell cytotoxicity in vitro, WR VV (wild type VV-*tk*+), WR-GFP (VV-*tk*^−^ GFP+), and WOTS-418 (VV-*tk*^−^, HSV-*tk_418m_*+ and Fluc^+^) were each used to infect three human and three murine cancer cell lines. All three viruses showed similar and remarkable cytotoxicity in HeLa and HCT 116 cells. In NCI-H460, WOTS-418 showed higher cytotoxicity than WR VV and WR-GFP. In addition, the murine cancer cell lines were resistant to all three viruses. However, WOTS-418 showed slightly higher cytotoxicity in the Renca cancer cell line. Overall, this study indicated that gene modification and incorporation of the *HSV-tk_418m_* transgene in WR VV backbone did not alter cytotoxic potency of this new recombinant OVV (Figure 1F, left panel). Interestingly, the virus yield (Figure 1F, right panel) for WOTS-418 ≤ WR-GFP but the cytotoxicity of WOTS-418 ≥ WR-GFP (Figure 1F, left panel), indicating WOTS-418 had a stronger cytolytic effect in the cancer cells tested in this in vitro study even with low WOTS-418 yield. However, the WR-GFP and WOTS-418 virus yield was not compared in normal cell line. Furthermore, the impact of WOTS-418 on cancer cells (HeLa S3, A549, Caki-1, HCT 116, NCI-H460, MDA-MB-231, MCF7, and 4T1) in terms of phase confluence was evaluated through Incucyte S3 imaging system. This data also supported the indication that WOTS-418 is capable of inducing a strong cytopathic effect from 24 h to over 120 h *p.i*. However, NCI-H460 and 4T1 were moderately resistant to WOTS-418 mediated cytolysis compared to the uninfected cells (Appendix B in Figure A3). Incucyte S3 raw video image analysis also showed virus mediated cytopathic effect, characterized by infected cells becoming round and cell detachment, in all tested cell lines (Appendix A, WOTS-418 infected NCI-H460 as a representative data). 

Furthermore, preclinical safety of WOTS-418 was also investigated in a non-tumor bearing syngeneic mouse model. In this study, intranasal delivery of low dose WR VV (1 × 10^5^ PFU and 1 × 10^7^ PFU) resulted in rapid body weight reduction (more than 30% by Day 7 *p.i.*) and early death (all dead by Day 11 *p.i.*) in treated mice. On the other hand, intranasal delivery of a similar (1 × 10^7^ PFU) or higher dose (5 × 10^8^ PFU) WOTS-418 did not have any effect on overall survival (Figure 1G, left panel) or mouse body weight (Figure 1G, right panel).

### 3.2. WOTS-418 Showed Enhanced Antiviral Potency to Purine Analogs (ACV and GCV) but No Antiviral Potency to Pyrimidine Analog

In the prodrug (purine and pyrimidine analog) antiviral potency experiment, BvdU (100 μM, pyrimidine analog) showed strong HSV-1 viral replication inhibition but WOTS-418 viral replication was unaffected by BvdU (Figure 2B). However, purine analogues, ACV and GCV at the same dose (100 μM) and condition showed statistically significant WOTS-418 viral plaque inhibition (Figure 2A). This data supported the previously reported pyrimidine sensitivity abolishment, characteristic of *HSV-tk_wt_*’s A167Y (p.Ala167Try) modification. Interestingly, when we compared between the two purine analogues (ACV vs. GCV), GCV treated groups (*p* = 0.01) showed significantly stronger WOTS-418 viral plaque inhibition than the ACV treated groups (*p* = 0.025). Therefore, from this point on, GCV was used to evaluate WOTS-418 viral replication control and suicidal effect.

### 3.3. WOTS-418 Replication and Cytolysis Were Attenuated in Normal Human Cell Line but Not in Cancer Cells

Incucyte S3 live imaging system (Figure 3A) and cell viability assessment by CCK-8 assay (Figure 3B) showed that wild type WR VV had more of a cytopathic effect and higher cytotoxicity (*p* = 0.003 at 0.1 PFU and *p* = 0.053 at 1 PFU) in MRC-5 normal human cell line than the WOTS-418. During virus replication assessment in normal human and cancer cell lines in vitro, WOTS-418 was used to infect MRC-5 and HeLa cell lines at the same dose 1 PFU/cell. Our result showed that the luciferase gene expression from WOTS-418 virus was significantly attenuated in normal human cell line when compared with HeLa cancer cell line (*p* = 0.003; Figure 3C). 

Next, the tumor selectivity of WOTS-418 was evaluated in HCT 116 tumor bearing mice and compared with WR VV. In brief, after seven days of systemic delivery (intraperitoneal) of either virus, the infectious virus particles in tumors and normal tissues (brain, liver, lung, spleen, kidney, ovary, testis, heart, and normal muscle) were quantified by vaccinia virus plaque assay. The vaccinia virus yield (*n* = 3) was normalized and calculated as Log_10_ PFU/g. A markedly higher titer of WR VV virus was seen in all the normal tissues (brain, liver, lung, spleen, kidney ovary, testis, heart, and normal muscle) tested in this experiment. WOTS-418 showed consistent replication attenuation in most of these normal tissues (Figure 3D). We found a high viral (WR VV and WOTS-418) load in the reproductive organs (testis and ovaries), however, WOTS-418 showed a relatively lower viral titer in those organs compared to WR VV. A minute level of WOTS-418 was detected in the brain, spleen, and kidney. Interestingly, WOTS-418 was not found in normal muscle tissue, liver, lung, or heart. Notably, WOTS-418 yield was relatively higher than that of wild type WR VV in the tumor. This study indicated that the deletion of VV *TK* and incorporation of modified *HSV-tk_418m_* did not impact tumor selectivity. Interestingly, this modification significantly attenuated the infectivity of WOTS-418 in all the normal tissues. 

### 3.4. WOTS-418 Showed Significant Antitumor Response in Human Solid Tumor Models

At first, OTS-412 and WOTS-418 antitumor potency was compared in an HCT 116 tumor bearing model. Following a single intratumoral dose (1 × 10^6^ PFU) of WOTS-418 (*p* < 0.0001, *n* = 7), there was a remarkable and identical antitumor response in this experimental model (Figure 4A). Furthermore, in a human renal (Caki-1) tumor bearing model, a single intratumoral dose (1 × 10^6^ PFU) of WOTS-418 resulted in a robust antitumor response (*p* = 0.008, *n* = 5) until Day 27 *p.i.* (Figure 4B). Further evaluation in saline-treated mice showed consistent tumor progression up to Day 85 *p.i.* Interestingly, most tumors in the WOTS-418 treated group showed remarkable tumor suppression up to the same time point (Figure 4C). In both experimental model, there was no significant change of body weight following WOTS-418 administration (Appendix B in Figure A4A,B). 

### 3.5. WOTS-418/GCV Combination Showed No Significant Suicidal Effect In Vitro Except in NCI-H460 Cancer Cells 

To evaluate the GCV mediated suicidal effect in various cancer cell lines, we performed CCK-8 assay following GCV treatment in WOTS-418 infected cells. Among the cancer cell lines tested, one out of five cell lines (NCI-H460: *p* = 0.004) showed significant GCV mediated cytotoxicity (Figure 5A). In a separate experiment, this robust GCV mediated suicidal effect was also observed in WOTS-418 infected NCI-H460 cells and was recorded by Incucyte S3 Imaging system until 90 h *p.i* (this data is also available as a movie file (.mp4) which is attached as Appendix A). Again, GCV antiviral potency in vitro was also evaluated by plaque assay using the same samples from Figure 5A. Our data indicated a statistically significant reduction in WOTS-418 plaques in the GCV treatment group (NCI-H460: *p* < 0.001; A498: *p* < 0.001; MDA-MB-231: *p* < 0.001; 4T1: *p* = 0.001) except MCF7 (*p* = 0.183) compared to the WOTS-418 only group (Figure 5B).

### 3.6. WOTS-418/GCV Combination Significantly Inhibit Virus Replication Not Only in Tumors but Also in Normal Tissues

In an animal model, we evaluated the effect of GCV co-treatment on WOTS-418 replication in the HCT 116 tumor bearing mouse model. WOTS-418 (1 × 10^6^ PFU) was injected intratumorally at day 0. From day 7 *p.i.*, GCV (50 mg/Kg) treatment was administered subcutaneously near the tumor region for five consecutive days (once daily, Day 7 *p.i.* to Day 11 *p.i.*) which resulted in a significant reduction of WOTS-418 viral DNA copies (*p* = 0.004) as determined by qPCR (Figure 6A).

In a separate in vivo study, we evaluated the impact of GCV on viral replication inhibition in normal organs and tumor. In this subcutaneous HCT 116 tumor bearing model, WOTS-418 (1 × 10^6^ PFU) was injected intraperitoneally at day 0, and from day 7 *p.i.,* GCV (*i.p.*; 50 mg/Kg) was administered for the next seven consecutive days (once daily, Day 7 *p.i.* to Day 13 *p.i.*). Interestingly, consecutive GCV treatment resulted in significant reduction of WOTS-418 viral DNA copies not only in tumors (*p* < 0.001) but also in circulation (*p* = 0.005), brain (*p* = 0.034) and other tissues such as kidney (*p* = 0.074) and spleen (*p* = 0.151) as shown in Figure 6B. These data imply that HSV-*tk*_418m_ has strong functional activity that can convert the inert GCV into a very cytotoxic form (GCV-*ppp*) which can ultimately control WOTS-418 virus replication (Appendix B in Figure A2).

## 4. Discussion

OTS-412 was our first recombinant oncolytic vaccinia virus engineered from Wyeth strain vaccinia virus, containing a truncated *HSV-tk* (36.1 kDa) [20]. OTS-412 demonstrated less cancer cytotoxicity and less viral replication inhibition compared to WOTS-418 in the presence of GCV. OVVs from WR backbone are generally more cytotoxic and have showed remarkable antitumor efficacy in preclinical studies [21]. However, in clinical studies, despite substantial tumor response to WR backbone based OVVs, more serious adverse events were also observed [3,22]. If undesirable viral replication can be controlled, there may be an opportunity to maximize clinical benefit of WR OVs. *HSV-tk* is differs from human cytosolic tk (H-*tk*) and *VV-tk* (Appendix B
Figure A1) in the fact that it can bind a broader range of nucleosides, but this trait can be abolished following a single amino acid substitution. A167Y *HSV-tk* mutation and functionality change in nucleoside selectivity were determined by predictive computer modeling, and the authors found that GCV potency against A167Y mutant *HSV-tk* was substantially increased and markedly reduced dThd competition [19]. A167Y *HSV-tk* mutant was recombined into WR to incorporate a fast and efficient viral replication inhibition method in the case that undesirable viral replication could lead to adverse effects in a clinical setting. Moreover, A167Y *HSV-tk* mutant does not phosphorylate pyrimidine nucleosides, which increases cancer selectivity and considerably minimizes viral replication in normal cells. This suggests that A167Y HSV-*tk* insertion in WOTS-418 provides cancer selectivity and additionally acts as strong chemosensitizer. Even though GCV/*HSV-tk_418m_* suicidal effect was significant in NCI-H460 cell line, we were not able to replicate similar results in other cancer cell lines; therefore, we believe that the overall benefit of GCV/*HSV-tk_418m_* suicidal effect is limited or requires further analyses. Our preclinical animal studies show significant tumor suppression by WOTS-418; therefore, we do not focus on the GCV suicidal modality at this time, but instead, its use in controlling undesirable viral replication in a clinical setting. This contrasts the existing trend for GCV use in cancer therapeutics, where GCV is not used to control oncolytic viral replication, but rather to enhance the therapeutic efficacy by inducing targeted cancer cytotoxicity; for example, the incorporation of wild type *HSV-tk* gene into adenovirus backbone for oncolytic adenovirus suicide gene therapy [23,24].

In this study, an amino acid substitution (A167Y)) selectively abolished HSV-*tk*_418m_’s pyrimidine affinity and increased purine analogs antiviral potency. Our data show GCV inhibition of WOTS-418 is significant both in vitro and in vivo. This contrasts with that of wild type *HSV-tk,* where potency of purine and purine analogs (e.g., GCV) is extremely low due to constant phosphorylation of highly expressed dTh in cancer cells. This was confirmed when the A167Y mutant HSV-*tk,* expressed in HSV-1, preferably phosphorylated GCV even in the presence of high levels of dThd [19]. In this study, we showed evidence that replication of WOTS-418 expressing *HSV-tk_418m_* was significantly blocked by GCV. 

During OTS-412 engineering, three different types of *HSV-tk* mutants were observed. Wild type *HSV-tk* could not be incorporated into Wyeth strain vaccinia virus backbone, perhaps due to transgene instability. More than 1500 isolated single clones were wild type *HSV-tk* positive right after HR, but in most the clones (99.8%), the transgene disappeared after multiple passages, and only around 0.2% persistently and permanently presented truncated versions of the transgene, even after multiple passages (Figure 1A, top panel). From these positive clones, three distinct patterns emerged, and genetic analysis showed that mutations in the *HSV-tk* transgene occurred by nucleotide-G insertion or deletion mostly in the 7G ‘hot spot’ area. Following this findings, silent mutations were made in these 7G and 6C locations (Figure 1B) in WOTS-418. Genetic stability of WOTS-418 was confirmed by restriction enzyme mapping, transgene sequencing, and western blotting before and after multiple passages of >100,000-fold amplification.

In this study, we developed a potent replication-controlled OVV, WOTS-418, using a more aggressive WR strain VV harboring *HSV-tk_418m_*. We confirmed that WOTS-418 expressed the full length *HSV-tk* which is identical to the wild type *HSV-tk* (HSV-1 virus) in terms of protein band size, while also expressing the selective abolishment of pyrimidine (BvdU) antiviral potency. 

WOTS-418 alone demonstrated its capacity to induce strong cytolysis against both murine and human cancer cell lines in vitro. For in vivo studies, two xenograft human solid tumor models (HCT 116 and Caki-1) were investigated. In both animal models, WOTS-418 effectively prevented tumor growth and induced tumor suppression. Furthermore, GCV treatment showed significantly higher viral replication inhibition in mice infected with WOTS-418 than in those infected with OTS-412. WOTS-418 alone will not be evaluated as single agent in clinical studies; therefore, to elucidate the therapeutic potency and safety of WOTS-418, we conducted multiple in vivo studies using different immune modulator treatments (e.g., myeloid modulators and/or immune checkpoint inhibitors) in combination with WOTS-418 (systemic administration) in a syngeneic mouse model, and dose escalation studies were conducted in a VX2 tumor bearing rabbit model. This data will be reported in our next articles.

Finally, a viral biodistribution study in the HCT 116 tumor bearing animal model revealed that systemic GCV treatment can notably control WOTS-418 replication not only in tumors but also in normal tissues and organs. These important results demonstrate that WOTS-418 could be a safer alternative to other WR based OVVs in clinical studies.

## 5. Conclusions

Our primary aim in this study was to engineer and characterize a new replication controllable OVV, WOTS-418, with high purine analogues (e.g., GCV) antiviral potency. WOTS-418 alone showed significant cytolysis capacity in multiple neoplastic cell lines and robust tumor response in two highly progressive human solid tumors. Furthermore, WOTS-418 showed significant replication attenuation in normal tissues compared to wild type WR VV, enhanced tumor selectivity in vivo and, most importantly, viral replication was significantly inhibited in tumors and normal tissues after GCV treatment indicating a rapid control of aberrant viral replication in emergency situation.

## 6. Patents

WOTS-418 was filed for the PCT application (PCT/KR2020/008472) at, 29 June 2020); the patent is pending. 

## Figures and Tables

**Figure 1 biomedicines-08-00426-f001:**
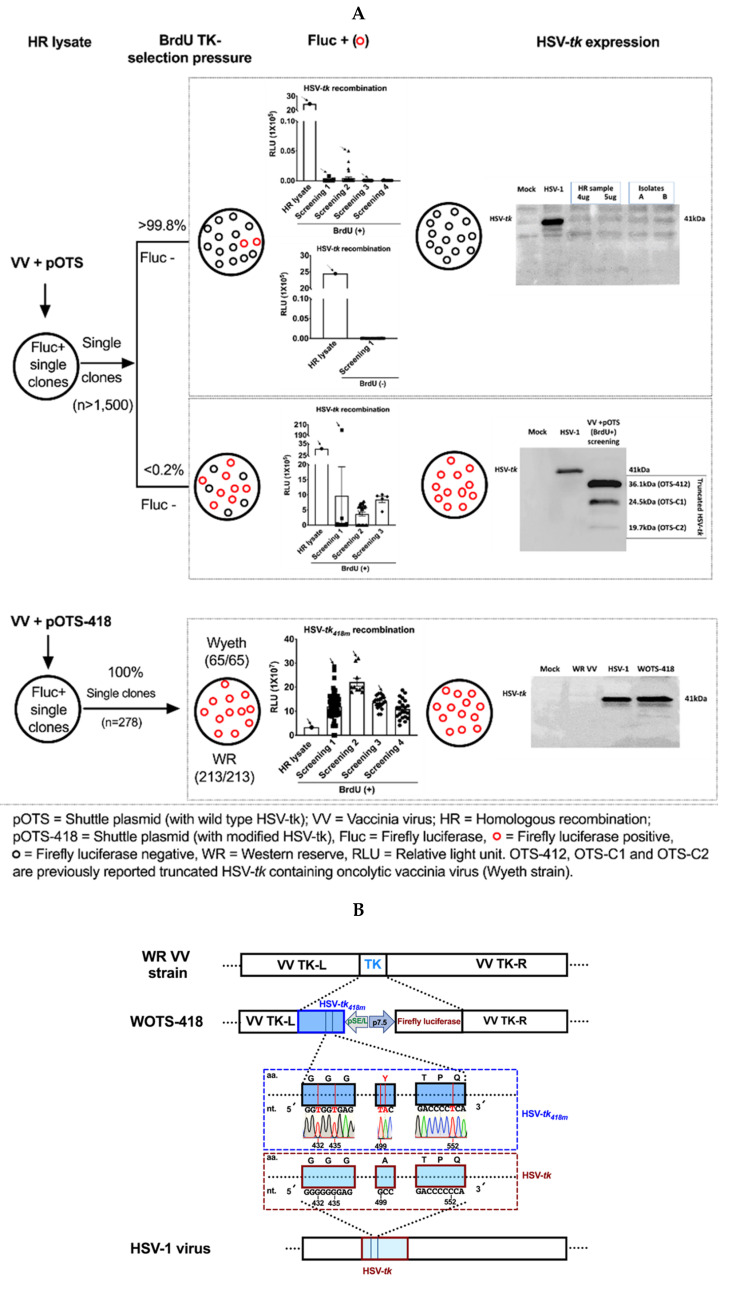
Rationale for engineering modified HSV-*tk* armed OVV, WOTS-418 and characterization. (**A**) Wild type *HSV-tk* transgene instability in VV backbone. Wild type *HSV-tk* or modified *HSV-tk_418m_* containing shuttle plasmid (pOTS or pOTS-418) was used for HR with VV. *Firefly luciferase* positive HR lysate were further screened with or without BrdU TK negative selection pressure. Isolated *firefly luciferase* positive singles plaques were further checked for *HSV-tk* expression by western blotting (detail screening and selection procedure described in the methods section). (**B**) WOTS-418 engineering map. Transgene insertion location in WR VV backbone is shown (top) and the diagram is showing the partial and representative location of the wild type HSV-tk DNA sequence (bottom). The middle panel is showing the representative modified HSV-*tk*_418m_ DNA sequence, indicating the location of altered nucleotides and DNA sequence confirmation. The promoters used for *HSV-tk_418m_* and *firefly luciferase* were pSE/L and p 7.5, respectively. (**C**) Monitoring long-term *HSV-tk_418m_* gene stability. To evaluate the long-term transgene stability, WOTS-418 virus samples from two different time points (first amplification and tenth amplification) were used. A549 cells were infected with WOTS-418 (0.1 PFU/cell), WR VV and HSV-1. 24 h post infection (*p.i.)*, cell pellets were used to measure HSV-*tk* expression by western blotting. WR VV: Wild type western reserve vaccinia virus (Negative control); HSV-1: Wild type HSV-1 virus (Positive control); Anti-HSV-1 thymidine Kinase; sc-28037, vN-20: SCBT, Dallas, TX, USA; Anti-GAPDH; MB001: Bioworld, Louis Park, MN, USA; Secondary antibody anti-goat; A50-101P: Bethyl, Montgomery, TX, USA; Secondary antibody anti-mouse; ADI-SAB-100-J: Enzo, Executive Blvd Farmingdale, NY, USA. Protein Marker: ExcelBand 3-Color PreStained Protein Markers, #2700 (Green BioResearch LLC, Baton Rouge, LA, USA), a mixture of blue, red, and green stained recombinant proteins (5 to 245 kDa), was used as size standards in western blotting. (**D**) WOTS-418 DNA stability: Samples from Figure 1C was also used for long-term DNA stability confirmation, which was evaluated by HindIII restriction digestion assay (detail procedure described in the methods section). Blue arrows indicate the alteration pattern of DNA band size after recombination. (**E**) GCV antiviral potency comparison between OTS-412 and WOTS-418: Quantitative analysis of viral DNA copies, measured in fold inhibition, was done following virus + GCV treatment and virus without GCV treatment. NCI-H460 and A549 cells were infected with either OTS-412 or WOTS-418 (0.1 PFU/cell) and co-treated with GCV (100 μM). 72 h *p.i.*, samples were harvested, and viral copy numbers were measured by quantitative polymerase chain reaction (qPCR). Data presented as mean ± SEM (*n* = 2) and compared using two-tailed Student’s *t*-test (*p*-value). (**F**) WR, WR-GFP, and WOTS-418 cytotoxicity comparison (left panel). Three human cancer cells (HeLa, NCI-H460, and HCT 116) and three murine cancer cells (Renca, CT-26.WT, and 4T1) were infected with WR, WR-GFP, WOTS-418 at a dose 0.1 PFU/cell for human cancer and 1 PFU/cell for murine cancer cells. 72 h *p.i.* samples were evaluated for cytotoxicity by CCK-8 assay. Data are presented as mean± SEM (*n* = 2). WR VV, WR-GFP, and WOTS-418 virus yield in various cancer cell lines (right panel): After cytotoxicity experiment, samples were harvested and subjected to three freeze and thaw cycles. Next, samples were sonicated with a cup sonicator (100% power for 30 s, 3 times with 1-min interval) and reinfected (same volume, 30 μL) in pre-seeded HeLa cells (10,000 cells/well) in 96-well plates and incubated at 37 °C with 5% CO_2_. 72 h *p.i.,* CCK-8 assay was performed to evaluate that viral cytotoxicity of the three viruses. Data are presented as mean ± SEM (*n* = 3). (**G**) WR vs. WOTS-418 safety assessment: Non-tumor bearing BALB/c syngeneic mice were treated intranasally with WR VV (1 × 10^5^ PFU and 1 × 10^7^ PFU) or WOTS-418 (1 × 10 ^7^ PFU and 5 × 10^8^ PFU). After WR VV or WOTS-418 treatment, survival and body weight were monitored. Survival curve demonstrating overall survival and *p* < 0.0001 was determined by log-rank (Mantel-Cox) test (*n* ≥ 3).

**Figure 2 biomedicines-08-00426-f002:**
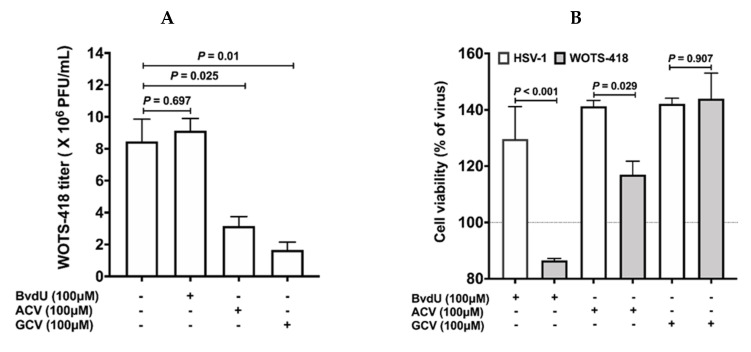
Evaluation of WOTS-418 (HSV*-tk_418_*) purine and pyrimidine analogs antiviral potency. (**A**) 143B cells (*tk* negative) were infected with WOTS-418 (0.1 PFU/cell) and co-treated with purine analogs (ACV and GCV), and pyrimidine analog (BvdU) at a dose 100 μM. 72 h *p.i.*, samples were harvested, and antiviral potency to each of these analogs was evaluated by vaccinia virus plaque titration assay in U-2 OS (**B**) 143B cells (*tk* negative) were infected with WOTS-418 (0.1 PFU/cell) or HSV-1 virus (positive control) and co-treated with purine analogs (ACV and GCV), and pyrimidine analog (BvdU) at a dose 100 μM. 72 h *p.i.*, samples were harvested and subjected to three freeze and thaw cycles. Next, samples were sonicated with a cup sonicator (100% power for 45 s, 3 times with 1-min interval) and reinfected (same volume, 100 μL) in pre-seeded HeLa cells (10,000 cells/well) in 96 well plates and incubated at 37 °C with 5% CO_2_. 72 h *p.i.,* cytotoxicity assay (CCK-8) was performed. Data presented as mean ± SEM (*n* ≥ 2) and compared using two-tailed Student’s *t*-test (*p*-value). ACV: Acyclovir, GCV: Ganciclovir and BvdU: (E)-5-(2-Bromovinyl)-2′-deoxyuridine.

**Figure 3 biomedicines-08-00426-f003:**
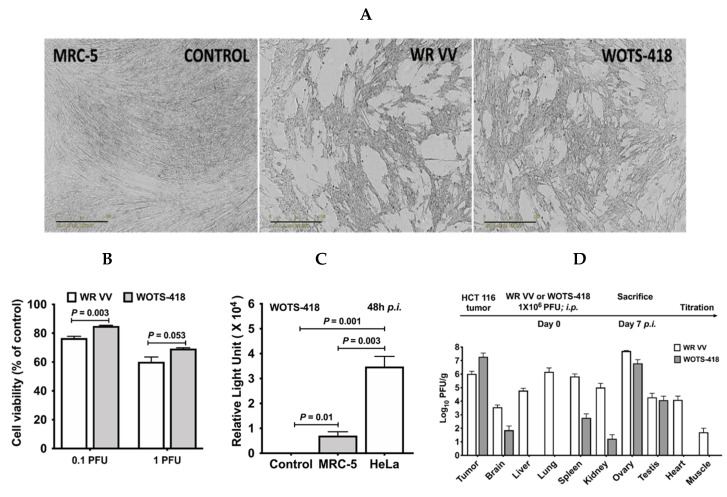
Tumor selectivity of WOTS-418. (**A**) Normal human cell line MRC-5 cells were seeded in 96-well plate at 3 × 10^4^ cells/well and infected with WR-VV or WOTS-418 (0.1 PFU/cell or 1 PFU/cell) to evaluate cytopathic effect and virus replication of WOTS-418. 48 h *p*.i., a phase contrast scan was performed using Incucyte S3 live cell imaging system (Scale bar: 400 μm). (**B**) Next, using the same samples from Figure 3A, cell viability was measured by cytotoxicity assay (CCK-8). Data presented as mean± SEM (*n* ≥ 3) and compared using two-tailed Student’s *t*-test (*p*-value). (**C**) To evaluate virus replication in MRC-5, a control cancer cell line, HeLa, was infected under the same conditions and after 48 h *p*.i., 20 μL supernatant was used to analyze the luciferase expression (E1500, Promega, WI, USA). Data presented as mean± SEM (*n* ≥ 3) and compared using two-tailed Student’s *t*-test (*p*-value). (**D**) HCT 116 tumor bearing male and female BALB/c nude mice were intraperitoneally injected with either WR VV or WOTS-418 at a dose 1 × 10^6^ PFU. 7 days *p*.i., mice were sacrificed, tumor and other normal organs (brain, liver, lung, spleen, kidney, ovary, testis, heart and muscle) were harvested aseptically. Weight of each organ was recorded. Samples were homogenized (Omni bead 24 rupture) and analyzed by titration (plaque assay) for vaccinia virus detection and quantification. Data are presented as Log_10_ PFU/gram and as a mean ± SEM (*n* = 3).

**Figure 4 biomedicines-08-00426-f004:**
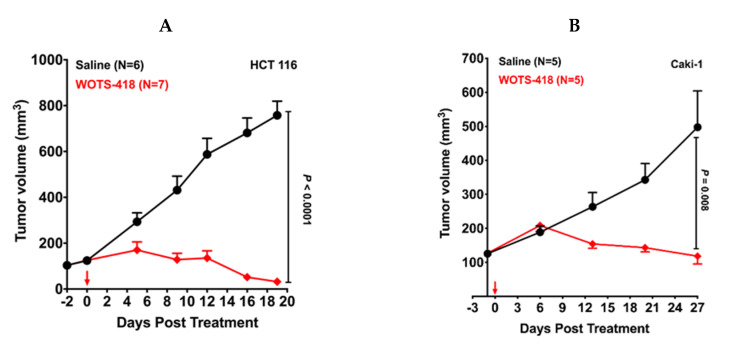
WOTS-418 antitumor potency in human colorectal (HCT 116) and renal (Caki-1) cancer models. (**A**) Human colorectal cancer (HCT 116) model: Antitumor effect following a single dose treatment of WOTS-418 was evaluated in HCT 116 human colorectal cancer model. Xenografts were established in the BALB/c athymic nude mice by injecting 5 × 10^6^ cells subcutaneously in the right flank of each mouse. After randomization and grouping, animals were treated intratumorally with either saline or WOTS-418, at a dose 1 × 10^6^ PFU on Day 0. Tumor and body weight (Appendix B in Figure A4A) were recorded at various time points and data are presented as mean ± SEM (*n* ≥ 6) and compared using two-tailed Student’s *t*-test (*p*-value). (**B**) Human renal cancer (Caki-1) model: Xenografts were established in the BALB/c athymic nude mice by injecting 1.75X10^6^ cells subcutaneously in the right flank of each mouse. After randomization and grouping, animals were treated intratumorally, with either saline or WOTS-418, at a dose1 × 10^6^ PFU at Day 0. Tumor and body weight (Appendix B in Figure A4B) were recorded at various time points until Day 27 *p.i.* and data are presented as mean ± SEM (*n* = 5) and compared using two-tailed Student’s *t*-test (*p-*value). (**C**) Treated mice from Caki-1 model were kept until Day 85 *p.i.* to monitor further tumor progression or regression. At 52 *p.i.* and day 85 *p.i.* images were captured. The day 85 *p.i.* photo was taken after sacrifice. Therefore, mouse skin color was faint. Confirmation of complete response was evaluated based on the absence of pulpable tumors.

**Figure 5 biomedicines-08-00426-f005:**
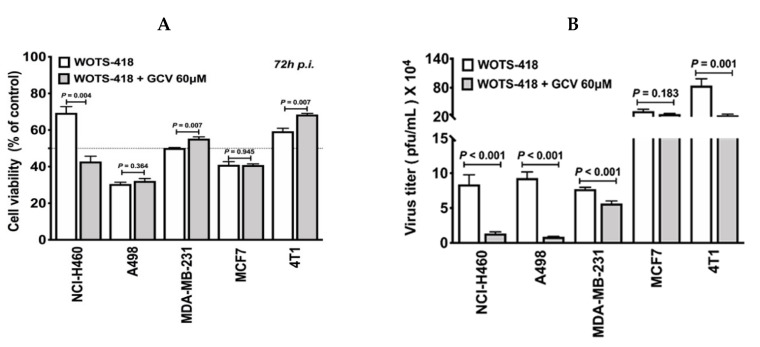
GCV suicidal effect on WOTS-418 infected cancer cells evaluation of viability and virus replication inhibition in vitro. (**A**) Impact of GCV on WOTS-418 infected cancer cells (NCI-H460, A498, MDA-MB-231, MCF7, and 4T1) was evaluated by cell viability assay. Cells (3 × 10^3^/well) were seeded in 96-cell plate and were infected with WOTS-418 at 1 PFU/cell the following day. After two hours, infection media was replaced with low dose GCV (60 μM) and incubated for 72 h at 37 °C with 5% CO_2_. Cell viability was measured by cell counting kit (CCK-8, Dojindo, Kumamoto, Japan). Data presented as a mean ± SEM (*n* = 3) and compared using two-tailed Student’s *t*-test (*p*-value). (**B**) Impact of low dose GCV (60 μM) on WOTS-418 plaque formation was monitored in multiple neoplastic cancer cell lines. Samples for Figure 5A were harvested, cycled through freezing and thawing three times, and sonicated with a cup sonicator (100% power for 30 s, three times with 1-min interval). Titration was performed in U2-OS cell line according to vaccinia virus titration protocol. Data presented as mean ± SEM (*n* ≥ 4) and compared using two-tailed Student’s *t*-test (*p-*value).

**Figure 6 biomedicines-08-00426-f006:**
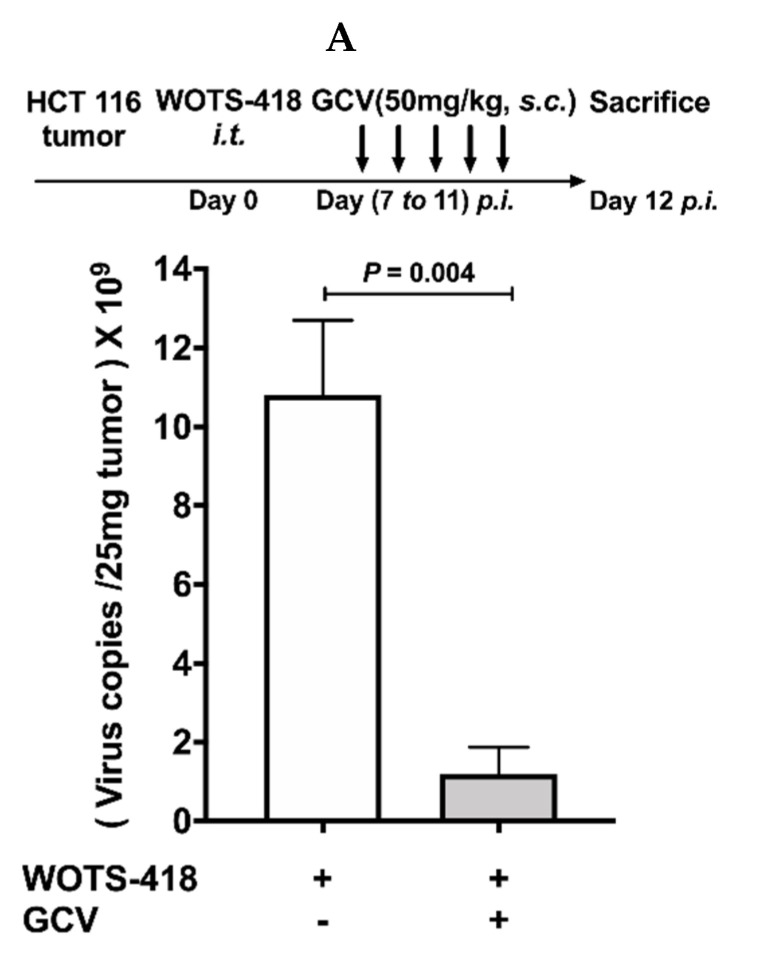
Dynamics of GCV antiviral potency for controlling WOTS-418 replication in animal model. (**A**) In vivo assessment of GCV antiviral potency to WOTS-418 replication was evaluated in an HCT 116 tumor bearing mouse model by qPCR analysis. Mice were injected subcutaneously with 5 × 10^6^ HCT 116 cells/mouse in the right flank of the BALB/c nu/nu mice. When palpable tumors were appeared, WOTS-418 was injected intratumorally at 1 × 10^6^ PFU dose and seven days post virus injection, subcutaneous GCV treatment (50 mg/Kg) performed daily for five times (D 7 *p.i.* to D 11 *p.i.*). Tumors harvested and homogenized using a bead rupture 24 (Omni international, Kennesaw, GA, USA) set to 3 cycles for 30 s at 4 m/s with 30 s intervals. WOTS-418 virus was quantified by qPCR. Data presented as mean ± SEM (*n* ≥ 2) and compared using two-tailed Student’s *t*-test (*p-*value). (**B**) In a separate in vivo study, GCV antiviral potency to WOTS-418 replication was evaluated in an HCT 116 tumor bearing mouse model and quantified by qPCR analysis. 5 × 10^6^ HCT 116 cells/mouse were injected subcutaneously in the right flank of the BALB/c nu/nu mice. When palpable tumors were appeared, WOTS-418 was injected intraperitoneally at 1 × 10^6^ PFU dose. 7 days following WOTS-418 injection, mice were given an intraperitoneal GCV treatment (50 mg/Kg) performed daily for seven times (D 7 *p.i.* to D 13 *p.i.*). Mice were sacrificed and blood samples were harvested in BD vacutainer blood collection tubes (BD, Franklin Lakes, NJ, USA). Other tissues (kidney, spleen, brain, and tumor) were harvested, weighed, and homogenized using bead rupture 24 (Omni international, Kennesaw, GA, USA) set to three cycles of 30 s at 4 m/s with 30 s intervals. WOTS-418 virus was quantified by qPCR. Data presented as mean ± SEM (*n* ≥ 2) and compared using two-tailed Student’s *t*-test (*p*-value).

**Table 1 biomedicines-08-00426-t001:** List of primers used in this study.

Primer List	Sequence	Purpose
HSV1-TK SF	5′-CCT CGT CGC AAT ATC GCA TTT T-3′	PCR
HSV1-TK SR	5’-CTC CAG CGG TTC CAT CTT C-3′	PCR
SEQ 1	5′-AGT TAG CCT CCC CCA TCT CC-3′	Sequencing
SEQ 2	5′-CGA CAG ATC TAG GCC TGG TA-3′	Sequencing
SEQ 3	5′-CCC TGC TGC AAC TTA CCT CC-3′	Sequencing
SEQ 4	5′-CTC CAG CGG TTC CAT CTT C-3′	Sequencing
E9L Forward	5′-CAA CTC TTA GCC GAA GCG TAT GAG-3′	qPCR
E9L Reverse	5′-GAA CAT TTT TGG CAG AGA GAG CC-3′	qPCR
Probe	5′-6-FAM -CAG GCT ACC AGT TCA A-MGB/NFQ-3′	qPCR

## Data Availability

The data supporting this this study are available on request from the corresponding author. However, these data are not publicly available due to privacy or ethical restrictions.

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
