# Peer review of "Engineering and Preclinical Evaluation of Western Reserve Oncolytic Vaccinia Virus Expressing A167Y Mutant Herpes Simplex Virus Thymidine Kinase"

_biomedicines, 2020, doi:10.3390/biomedicines8100426_

Round 1

Reviewer 1 Report

The authors constructed the genetically engineered vaccinia virus WOTS-418 in which the tk gene was replaced by the HSV-tk gene with A167Y mutation from the WRvv. WOTS-418 was less virulent than the parental WRvv, but its cytotoxicity to cancer cells was almost similar to that of WRvv. The replication of WOTS-418 was inhibited by GCV, but not BvdU, indicating the sensitivity of WOTS-418 to anti-HSV drugs.

In the introduction, the authors stated that tk-deleted VV have been developed to increase cancer selectivity using methods to substantially attenuate viral replication in normal cells, but these methods can also significantly decrease cytotoxic potency in cancer cells, which may lead to relatively poor clinical outcomes. If the authors wish to demonstrate that the introduction of HSV-tk418m enhances the cytotoxicity of tk-deleted VV, they should examine the virus yield along with the cytotoxicity of WR-GFP as well as WOTS-418 in normal cells and various types of cancer cells and present the results in Figures 3B and 3C.

GCV did not enhance the cytotoxicity of WOTS-418 in cell lines other than NCI-H460. Therefore, the benefit of replacing the VV-tk gene by HSV-tk418m seems to be that the use of anti-HSV drugs prevents the spread of uncontrollable viruses in the body. Why hasn’t the effect of ACV rather than GCV on WOTS-418 replication been extensively investigated?

Many genetically engineered VVs have been developed. For example, GL-ONC1 (GLV-1h68), which has multiple genes modified, was developed from WRvv and is safely administered to cancer patients. It is needed to explain how WOTS-418 is superior to these oncolytic VVs in clinical practice.

The advantage of VV as an oncolytic virus over oncolytic HSV-1 such as T-vec is that it can be used intravenously to infect metastatic lesions. The effect of intravenous administration of WOTS-418 on cancers is not shown in this paper.

Page 7, lines 4-30: Section 3.1 is not required for the results in this paper.

In Figure 6, GCV treatment was performed from day 7 to day 11 (or 13) after WOTS-418 injection (day 0). Why did GCV treatment begin 2 h after WOTS-418 infection and end 72 h later in cell culture (Figure 5)?

In Figure 6A, WOTS-418 was injected i.t., but in the legend it was injected intraperitoneally. This needs to be rectified.  

Reviewer 2 Report

The authors present a strategy to improve the safety of oncolytic vaccinia virus by means of incorportaing a mutan herpes simplex virus thymidine kinase gene that showed purine exclusive selectivity. The work is fine regarding characterization of the new versión of the OVV. In vivo results need additional clarification:

1.- What cells are the target in brains, ovarian and testis? This information would help in preventing toxicities.

2.- Did the authors collect any data on toxicities in the mice treated with the OVV?

3.- Did the authors attempt testing safety and efficacy in vivo? Given that the aim is to improve the OVV medicine for therapy it is very important to show that enhance safety is not gain through diminishing efficacy.

4.- Did the authors try to test the in vivo immune response against this new OVV versión?

Round 2

Reviewer 1 Report

The author responds appropriately to the points raised by the reviewers.

Reviewer 2 Report

The authors addressed each question.